# Improvement of renal function after transcatheter aortic valve replacement in patients with chronic kidney disease

Michel V. Lemes da Silva[1,2], Antonio C. B. Nunes Filho[1] *, Vitor E. E. Rosa[1,2], Adriano Caixeta[1], Pedro A. Lemos Neto[1], Henrique B. Ribeiro[2], Breno O. Almeida[1], José Mariani, Jr[1,2], Carlos M. Campos[2,3], Alexandre A. C. Abizaid[2], José A. Mangione[4], Roney O. Sampaio[2], Paulo Caramori[5], Rogério Sarmento-Leite[6], Flávio Tarasoutchi[1,2], Marcelo Franken[1], Fábio S. de Brito, Jr[2]

1 Department of Cardiology, Hospital Israelita Albert Einstein (Albert Einstein Hospital), Sao Paulo, Brazil, 2 Department of Cardiology, Heart Institute (InCor), Clinical Hospital, Faculty of Medicine, University of Sao Paulo, Sao Paulo, Brazil, 3 Department of Cardiology, Instituto Prevent Senior, Sao Paulo, Brazil, 4 Department of Interventional Cardiology, Hospital Beneficiência Portuguesa, Sao Paulo, Brazil, 5 Department of Interventional Cardiology, Hospital São Lucas – PUCRS, Porto Alegre, Brazil, 6 Department of Interventional Cardiology, Instituto de Cardiologia do Rio Grande do Sul, Porto Alegre, Brazil

* antonio.filho@einstein.br

**Data Availability Statement:** All relevant data are within the manuscript and its Supporting information files.

## Abstract

### Background

Chronic kidney disease is commonly found in patients with aortic stenosis (AS) undergoing transcatheter aortic valve replacement (TAVR) and has marked impact in their prognosis. It has been shown however that TAVR may improve renal function by alleviating the hemodynamic barrier imposed by AS. Nevertheless, the predictors of and clinical consequences of renal function improvement are not well established.

Our aim was to assess the predictors of improvement of renal function after TAVR.

### Methods

The present work is an analysis of the Brazilian Registry of TAVR, a national non-randomized prospective study with 22 Brazilian centers. Patients with baseline renal dysfunction (estimated glomerular filtration rate [eGFR] < 60mL/min/1.73m$^2$) were stratified according to renal function after TAVR: increase >10% in eGFR were classified as TAVR induced renal function improvement (TIRFI); decrease > 10% in eGFR were classified as acute kidney injury (AKI) and stable renal function (neither criteria).

### Results

A total of 819 consecutive patients with symptomatic severe AS were included. Of these, baseline renal dysfunction (estimated glomerular filtration rate [eGFR] < 60mL/min/1.73m$^2$) was present in 577 (70%) patients. Considering variance in renal function between baseline and at discharge after TAVR procedure, TIRFI was seen in 197 (34.1%) patients, AKI in 203 (35.2%), and stable renal function in 177 (30.7%).

**Funding:** Funding sources were provided by Sociedade Brasileira de Hemodicâmica e Cardiologia Intervencionista (SBHCI) for data collection and analysis for the Brazilian TAVR Registry.

**Competing interests:** I have read the journal's policy and the authors of this manuscript have the following competing interests: Dr. de Brito Jr. and Dr. Mangione are proctors for Edwards Lifescience and Medtronic. Dr. Ribeiro is proctor and consultant for Edwards Lifescience, Medtronic and Boston Scientific. Dr. Caramori is proctor for Medtronic. All other authors have reported that they have no relationships relevant to the contents of this paper to disclose. This does not alter our adherence to PLOS ONE policies on sharing data and materials.

The independent predictors of TIRFI were: absence of coronary artery disease (OR: 0.69; 95% CI 0.48–0.98; P = 0.039) and lower baseline eGFR (OR: 0.98; 95% CI 0.97–1.00; P = 0.039). There was no significant difference in 30-day and 1-year all-cause mortality between patients with stable renal function or TIRFI. Nonetheless, individuals that had AKI after TAVR presented higher mortality compared with TIRFI and stable renal function groups (29.3% vs. 15.4% vs. 9.5%, respectively; p < 0.001).

## Conclusions

TIRFI was frequently found among baseline impaired renal function individuals but was not associated with improved 1-year outcomes.

## Introduction

Transcatheter aortic valve replacement (TAVR) is a well-established treatment for patients who cannot undergo surgery and those with intermediate to high surgical risk with symptomatic severe aortic stenosis [1–6]. More recently, TAVR indications have also been expanded for patients at low surgical risk, as well as for dysfunctional bioprosthesis [7, 8].

Among the patients currently undergoing TAVR, a high prevalence of non-cardiac comorbidities are frequently observed, including chronic kidney disease (CKD) that ranged from 52% to 72% in prior studies [9–11]. Moreover, renal function impairment at baseline also denotes worse clinical outcomes following TAVR, including higher mortality rates [10–14], particularly when acute kidney injury (AKI) after TAVR ensues [9, 15–17]. Of note, some patients may also experience renal function improvement after TAVR, regardless of the baseline renal condition, and also despite the various procedural factors that could jeopardize renal function such as hypotension during rapid pacing, use of iodinated contrast media, bleeding and athero-emboli. Yet, such adverse effects may be mitigated by the beneficial effects after the AS relief leading to improvement in cardiac output and better renal perfusion [9, 15, 16].

However, there is divergence in recent studies about the effects of the improvement in renal function after TAVR regarding better outcomes, and its real role still needs more investigation [9, 16].

This study aims to determine the predictors and prognosis of renal function improvement after TAVR using data from the Brazilian TAVR Registry.

## Methods

### Study population

From January 2008 to January 2015, 819 consecutive patients with symptomatic severe AS underwent TAVR and were included in the Brazilian TAVR Registry [18], which is a non-randomized prospective registry, including 22 Brazilian centers. Patients with baseline impaired renal function, defined as estimated glomerular filtration rate (eGFR) <60 mL/min/1.73m$^2$, were selected. eGFR was calculated using Chronic Kidney Disease Epidemiology formula (CKD-EPI) [19], expressed by GFR = 141 $*$ min(Scr/$\kappa$,1)$^{\alpha}$ $*$ max(Scr/$\kappa$, 1)$^{-1.209}$ $*$ 0.993$^{Age}$ $*$ 1.018 [if female] $*$ 1.159 [if black], which Scr is serum creatinine (mg/dL), $\kappa$ is 0.7 for females and 0.9 for males, $\alpha$ is -0.329 for females and -0.411 for males, min indicates the minimum of Scr/$\kappa$ or 1, and max indicates the maximum of Scr/$\kappa$ or 1. Also, 25 patients who died within the first 24 hours and 60 patients with missing data (4 patients without baseline eGFR, 55

patients without eGFR at discharge and 1 patient without eGFR at both moments) were excluded, remaining 577 patients who were included in the study ([Fig 1]). The study protocol was conducted in accordance with the Declaration of Helsinki and was approved by each institution's ethics committee under protocol number 05676012.4.1001.00701, and all participants had provided informed consent. The follow-up was performed by phone calls at 1 month, 1 year and then annually.

## Procedure

Patients with symptomatic severe AS (aortic valve area $< 1.0 cm^2$ and mean aortic valve gradient of $\geq$ 40 mmHg or peak aortic jet velocity of $\geq$ 4.0 m/s) were submitted to TAVR with self-expandable CoreValve prosthesis (Medtronic, Minneapolis, MN, USA), balloon-expandable Sapien XT prosthesis (Edwards Lifesciences, Irvine, CA, USA) or balloon-expandable Inovare prosthesis (Braile Biomédica, São José do Rio Preto, SP, Brazil). TAVR procedures were performed according to standard techniques. Aspirin (100 mg/day) and clopidogrel (300 mg loading dose and 75 mg/day) were administered to patients for at least 1 month. The preferable access was the transfemoral approach. When it was not feasible, transapical or transarterial approaches (transubclavian, direct transaortic, transapical or transcarotid) were used according to the preference of the Heart Team. Clinical, procedural and echocardiographic data were prospectively gathered into a pre-set TAVR database. Outcomes were defined according to the Valve Academic Research Consortium 2 (VARC-2) [20]. All outcomes and adverse events were adjudicated by an independent committee.

## Group definition

Serum creatinine was collected at baseline and daily after the procedure until discharge. The first creatinine (baseline) was collected on the day or the day before TAVR procedure. The second creatinine used to calculate variation on renal function was at discharge. The median length of stay was 7 days (1–368) and the mean hospitalization period was 13 days. The CKD-EPI formula was used to calculate the eGFR at baseline and at discharge after TAVR procedure. Patients were categorized according to renal function variation after TAVR: increase >10% in eGFR were classified as TAVR induced renal function improvement (TIRFI); decrease > 10% in eGFR were classified as AKI; and stable renal function (neither criteria) [9, 15].

## Statistical analysis

Continuous variables were reported as mean ± standard deviation, and the comparison between the 3 groups was performed using the ANOVA test, except for contrast media volume which Kruskal-Wallis test was used. Post-hoc analysis was performed using Tukey's test. Categorical variables were reported as frequencies and percentages, and were compared using the Pearson chi-square test. Cox proportional hazards models were used to test the impact of TIRFI and AKI on all-cause death and cardiovascular mortality, and the models were adjusted for age, male sex, New York Heart Association functional class III/IV, diabetes, hypertension, chronic obstructive lung disease, pulmonary hypertension, coronary artery disease, peripheral vascular disease, previous coronary artery bypass grafting, STS score, baseline eGFR, use of diuretics, Angiotensin-converting enzyme inhibitors or Angiotensin II receptor blockers, beta-blockers, statin, left ventricular ejection fraction, aortic valve area, mean aortic valve gradient, contrast media volume, procedural access (except transfemoral), Inovare prosthesis, AKI and TIRFI, myocardial infarction, stroke/transient ischemic attack, major or life-threatening bleeding, major vascular complication, new left bundle branch block, atrioventricular

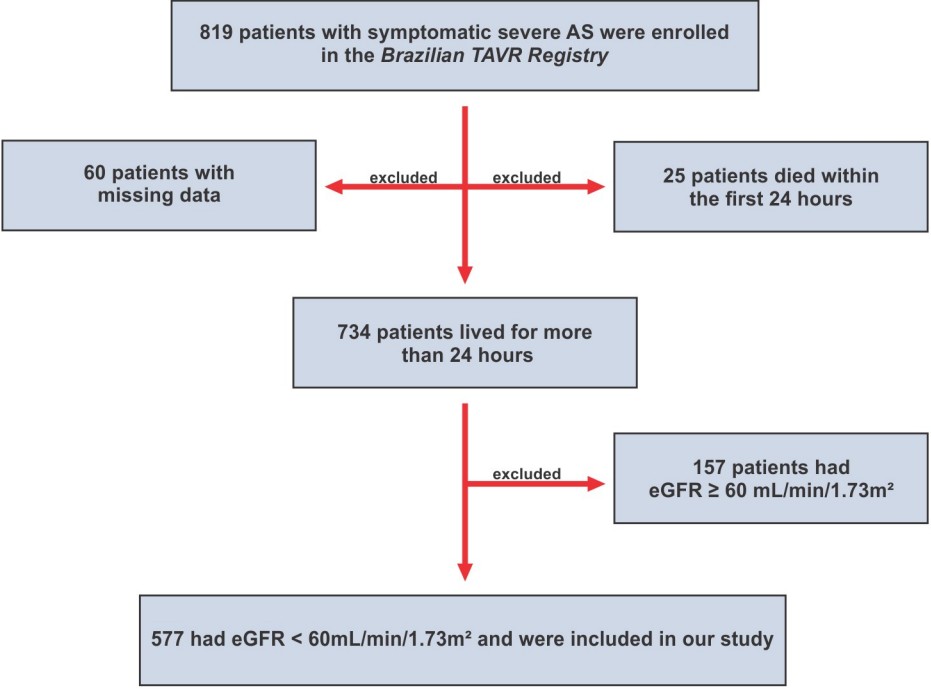

**Fig 1. Study flowchart.** Selection of the study population. Abbreviations: AS indicates aortic stenosis; eGFR, estimated glomerular filtration rate; TAVR, transcatheter aortic valve replacement.

block, valve malpositioning and new pacemaker. Kaplan-Meier survival curves were created comparing TIRFI, stable renal function and AKI groups and the outcomes were compared using a log-rank test. A stepwise logistic regression analysis including all variables with a P value < 0.2 in the univariate analysis was used to identify the predictors of TIRFI and AKI after the procedure until discharge. The models were adjusted for age, male sex, New York Heart Association functional class III/IV, diabetes, hypertension, chronic obstructive lung disease, pulmonary hypertension, coronary artery disease, peripheral vascular disease, previous coronary artery bypass grafting, STS score, baseline eGFR, use of diuretics, Angiotensin-converting enzyme inhibitors or Angiotensin II receptor blockers, beta-blockers, statin, left ventricular ejection fraction, aortic valve area, mean aortic valve gradient, contrast media volume, procedural access (except transfemoral), Inovare prosthesis. All statistical tests were 2-sided, and the criteria for statistical significance was P < 0.05. All statistical analyses were performed using SPSS statistical software version 20.0 (IBM, Armonk, New York, USA).

## Results

### Baseline clinical characteristics

Baseline characteristics of the study population are presented in Table 1. A total of 577 (70%) patients had CKD and were the population of the present study. The overall median length of stay was 7 days (1–368) and the mean hospitalization period was 13 days. After TAVR, 197 (34.1%) patients had TIRFI, 177 (30.7%) maintained stable renal function and 203 (35.2%) had AKI. The overall mean age among CKD patients was 81.3 ± 6.8 years, 56.2% were male, 31.7% had diabetes, 74.5% had hypertension and the mean baseline eGFR was 39.1 ± 12.2 ml/min/1.73m$^2$. The mean STS score was 10.6 ± 7.9% and the preferable access site was transfemoral

**Table 1. Baseline and procedural characteristics of the study population.**

| | TIRFI Group | Stable Renal Function Group | AKI Group | Overall | |
|---|---|---|---|---|---|
| | (n = 197) | (n = 177) | (n = 203) | (n = 577) | P value [a] |
| **Clinical data** | | | | | |
| Age, years | 81.3 ± 7.1 | 81.7 ± 7.2 | 82.6 ± 6.2 | 81.3 ± 6.8 | 0.14 |
| Male sex | 105 (53.3) | 111 (62.7) | 108 (53.2) | 324 (56.2) | 0.10 |
| NYHA class III/IV | 166 (84.3) | 146 (82.5) | 169 (83.3) | 481 (83.4) | 0.89 |
| Diabetes | 55 (27.9) | 59 (33.3) | 69 (34.0) | 183 (31.7) | 0.36 |
| Hypertension | 149 (75.6) | 117 (66.1) | 164 (80.8) | 430 (74.5) | **0.004** [b] |
| COPD | 38 (19.3) | 36 (20.3) | 37 (18.8) | 111 (19.2) | 0.87 |
| Pulmonary hypertension | 41 (20.8) | 50 (22.8) | 47 (23.2) | 138 (23.9) | 0.23 |
| CAD | 110 (55.8) | 118 (66.7) | 123 (60.6) | 351 (60.8) | 0.10 |
| Peripheral vascular disease | 35 (17.8) | 26 (14.7) | 35 (17.2) | 96 (16.6) | 0.69 |
| Previous CABG | 40 (20.3) | 44 (24.9) | 36 (17.7) | 120 (20.8) | 0.22 |
| STS score, % | 10.9 ± 8.4 | 10.2 ± 7.2 | 10.6 ± 8.1 | 10.6 ± 7.9 | 0.67 |
| eGFR, mL/min/1.73m$^2$ | 37.3 ± 12.5 | 39.6 ± 11.7 | 40.2 ± 12.3 | 39.1 ± 12.2 | **0.043** [c] |
| Diuretics | 130 (66.0) | 105 (59.3) | 134 (66.0) | 369 (64.0) | 0.30 |
| ACE inhibitors or ARB | 109 (55.3) | 80 (45.2) | 108 (53.2) | 297 (51.5) | 0.12 |
| Beta-blockers | 77 (39.1) | 68 (38.4) | 75 (36.9) | 220 (38.1) | 0.90 |
| Statin | 117 (59.4) | 101 (57.1) | 128 (63.1) | 346 (60.0) | 0.48 |
| **Echocardiographic data** | | | | | |
| LVEF, % | 56.8 ± 15.5 | 57.2 ± 16.0 | 58.4 ± 15.3 | 57.5 ± 15.6 | 0.54 |
| Mean transaortic gradient, mmHg | 49.8 ± 16.5 | 46.5 ± 15.9 | 48.1 ± 15.8 | 48.2 ± 16.1 | 0.15 |
| AVA, cm$^2$ | 0.68 ± 0.22 | 0.66 ± 0.17 | 0.66 ± 0.17 | 0.67 ± 0.19 | 0.55 |
| **Procedural data** | | | | | |
| Access site | | | | | 0.12 |
| Transfemoral approach | 188 (95.4) | 167 (94.4) | 184 (90.6) | 539 (93.4) | |
| Other | 9 (4.6) | 10 (5.6) | 19 (9.4) | 38 (6.6) | |
| Prosthesis type | | | | | 0.80 |
| Corevalve | 147 (74.6) | 128 (72.3) | 151 (74.4) | 426 (73.8) | |
| Sapien XT | 46 (23.4) | 43 (24.3) | 44 (21.6) | 133 (23.1) | |
| Inovare | 4 (2.0) | 6 (3.4) | 8 (3.9) | 18 (3.1) | |
| Contrast media volume, mL | 188 ± 120 | 186 ± 101 | 181 ± 91 | 185 ± 105 | 0.86 |
| Period of TAVR procedure | | | | | 0.15 |
| T1 (2008–2010) | 35 (17.8) | 23 (13.0) | 42 (20.7) | 100 (17.3) | |
| T2 (2011–2013) | 112 (56.9) | 118 (66.7) | 122 (60.1) | 352 (61.0) | |
| T3 (2014–2015) | 50 (25.4) | 36 (20.3) | 39 (19.2) | 125 (21.7) | |

Values are n (%) or mean (± SD).

[a] Represents significant statistical difference (P<0.05) between TIRF, stable and AKI groups.

[b] Significant difference (P<0.05) between to stable and AKI groups.

[c] Significant difference (P<0.05) between TIRFI and AKI groups.

Abbreviations: ACE, angiotensin-converting enzyme; AKI, acute kidney injury; ARB, angiotensin receptor blocker; AVA, aortic valve area; CABG, coronary artery bypass graft; CAD, coronary artery disease; COPD, chronic obstructive pulmonary disease; eGFR, estimated glomerular filtration rate; LVEF, left ventricular ejection fraction; NYHA, New York Heart Association; STS, Society of Thoracic Surgeons; T, tertile; TIRFI, TAVR induced renal function improvement.

(93.4%). Comparing baseline characteristics between the 3 groups, the mean eGFR was 37.3 ± 12.5 ml/min/1.73m$^2$ in the TIRFI group, 39.6 ± 11.7 ml/min/1.73m$^2$ in the stable renal function group and 40.2 ± 12.3 ml/min/1.73m$^2$ in the AKI group, with significant statistical difference between the 3 groups (P = 0.043). This difference was related to TIRFI and AKI

groups (post-hoc analysis, P = 0.044). Also, hypertension was present among 75.6% patients in TIRFI group, 66.1% in stable renal function group and 80.8% in AKI group, with significant difference between the 3 groups (P = 0.004). This difference was related to stable renal function and AKI groups (post-hoc analysis, P = 0.003). There was no difference related to contrast media volume and other baseline clinical, echocardiographic and procedural characteristics between the 3 groups. In order to evaluate the impact of the extended period of enrollment, we divided our study population into tertiles according to the year they underwent TAVR: 100 (17.3%) patients were submitted to TAVR from 2008 to 2010 (T1), 352 (61.0%) patients from 2011 to 2013 (T2), and 125 (21.7%) from 2014 to 2015 (T3), with no statistical differences between the 3 tertiles (p = 0.15).

## Procedural and clinical outcomes

Procedural and clinical outcomes are described in Table 2. The occurrence of valve malpositioning was higher in the AKI group compared with TIRFI and stable renal function groups (8.4% vs. 2.0% vs. 2.8%, respectively; P = 0.004). Also, major or life-threatening bleeding was higher in AKI group (20.2% vs. 6.7% vs. 6.3%, respectively; P < 0.001) as well as major vascular complication (10.5% vs. 3.6% vs. 4.0%, respectively; P = 0.006).

## Impact of TIRFI and AKI on short-term outcomes

At 30 days, patients in AKI group had higher rates of all-cause mortality compared with TIRFI and stable renal function groups (13.0% vs. 2.0% vs. 1.2%, respectively; P < 0.001), and also had higher rates of cardiovascular death (10.6% vs. 1.5% vs. 0.6%, respectively; P < 0.001). There was no significant difference regarding all stroke/transient ischemic attack and myocardial infarction and there was no significant difference in outcomes between TIRFI and stable renal function groups.

## Follow-up at 1 year and loss of follow-up

Of the 577 patients included in our study, 157 (27.2%) patients lost follow-up or did not complete 1 year of follow-up until the time for data collection for the Brazilian TAVR Registry. However, the median follow-up of the patients included in the present study was 385 [162–742] days. Data comparing the baseline characteristics of the patients with loss of follow-up and those with complete 1-year follow-up are shown as supplemental data (S1 Table).

## Impact of TIRFI and AKI on long-term outcomes

At 1 year, AKI group remained with higher rates of all-cause mortality compared with TIRFI and stable renal function groups (29.3% vs. 15.4% vs. 9.5%, respectively; P < 0.001), and had higher rates of cardiovascular death (21.0% vs. 6.0% vs. 4.5%, respectively; P < 0.001). There was still no difference regarding stroke/transient ischemic attack and myocardial infarction and there was no significant difference in outcomes between TIRFI and stable renal function groups group. Kapplan-Meier curves for 1-year all-cause mortality are represented in Fig 2.

## Predictors of TIRFI and AKI

Multivariate analysis of TIRFI and AKI are shown in Tables 3 and 4, respectively. The absence of coronary artery disease was an independent predictor of TIRFI (OR: 0.69; 95% CI 0.48–0.98; P = 0.039) as well as lower baseline eGFR (OR: 0.98; 95% CI 0.97–1.00; P = 0.008). Otherwise, independent predictors of AKI were age (HR: 1.03; 95% CI, 1.00–1.05; P = 0.033), hypertension (HR: 1.69; 95% CI, 1.11–2.57; P = 0.014), non-transfemoral access sites (HR: 2.07; 95%

**Table 2. Procedural, 30-day and 1-year outcomes.**

| | TIRFI Group | Stable Renal Function Group | AKI Group | |
|---|---|---|---|---|
| | (n = 197) | (n = 177) | (n = 203) | P value [a] |
| **Procedural variables** | | | | |
| Valve malpositioning | 4 (2.0) | 5 (2.8) | 17 (8.4) | **0.004**[(b,c)] |
| Major of life-threatening bleeding | 13 (6.7) | 11 (6.3) | 41 (20.2) | < **0.001**[(b,c)] |
| Major vascular complication | 7 (3.6) | 7 (4.0) | 21 (10.5) | **0.006**[(b,c)] |
| **30-day outcomes** | | | | |
| All-cause death | 4 (2.0) | 2 (1.2) | 26 (13.0) | <**0.001**[(b,c)] |
| Cardiovascular death | 3 (1.5) | 1 (0.6) | 22 (10.6) | <**0.001**[(b,c)] |
| All stroke/TIA | 5 (2.6) | 6 (3.4) | 11 (5.6) | 0.28 |
| Myocardial infarction | - | - | 1 (0.5) | 0.39 |
| New pacemaker | 38 (19.5) | 41 (23.3) | 48 (23.5) | 0.56 |
| New persistent LBBB | 51 (25.9) | 40 (22.6) | 63 (31.1) | 0.15 |
| **1-year outcomes** | | | | |
| All-cause death | 30 (15.4) | 17 (9.5) | 59 (29.3) | <**0.001**[(b,c)] |
| Cardiovascular death | 12 (6.0) | 8 (4.5) | 43 (21.0) | <**0.001**[(b,c)] |
| All stroke/TIA | 5 (2.6) | 6 (3.4) | 10 (5.6) | 0.21 |
| Myocardial infarction | 2 (0.8) | - | 4 (1.8) | 0.16 |
| New pacemaker | 45 (23.0) | 48 (27.4) | 51 (25.2) | 0.12 |
| New persistent LBBB | 52 (26.5) | 41 (23.2) | 67 (32.9) | 0.12 |

[a] Represents significant statistical difference (P<0.05) between TIRF, stable and AKI groups.

[b] Significant difference (P<0.05) between stable and AKI groups.

[c] Significant difference (P<0.05) between TIRFI and AKI groups.

Values are n (%).

Abbreviations: AKI, acute kidney injury; LBBB, left bundle branch block; TIA, transient ischemic attack; TIRFI, TAVR induced renal function improvement.

CI, 1.06–4.07; P = 0.034) and higher baseline eGFR (HR: 1.01; 95% CI, 1.00–1.02; P = 0.053). The multivariable analysis of TIRFI and AKI including the procedure complications after TAVR in the model are shown in S2 and S3 Tables).

## Predictors of 1-year all-cause mortality

Univariable and multivariable predictors of 1-year all-cause mortality are shown in Table 5. The presence of AKI was associated with an increase in 1-year all-cause mortality (HR: 3.42; 95% CI, 1.87–6.23; P < 0.001), as well as chronic obstructive pulmonary disease (HR: 1.72; 95% CI, 1.06–2.79; P = 0.028), baseline eGFR (HR 0.72; 95% CI, 0.61–0.86. P < 0.001) and stroke/transient ischemic attack (HR: 3.08; 95% CI, 1.74–5.44; P < 0.001).

## Discussion

The main findings of the present study can be summarized as: (1) CKD was extremely prevalent among our population, corresponding to 70% of the patients; (2) TIRFI was frequently found after TAVR; (3) predictors of TIRFI were the absence of coronary artery disease and lower eGFR at baseline; (4) patients with AKI after TAVR had higher all-cause mortality and cardiovascular death at 30 days and 1 year; (5) there was no differences comparing 30-day and 1-year outcomes between TIRFI and stable renal function groups.

Patients with severe AS often have multiple comorbidities, and impaired renal function is associated with poor outcomes with surgical aortic valve replacement [21] and TAVR

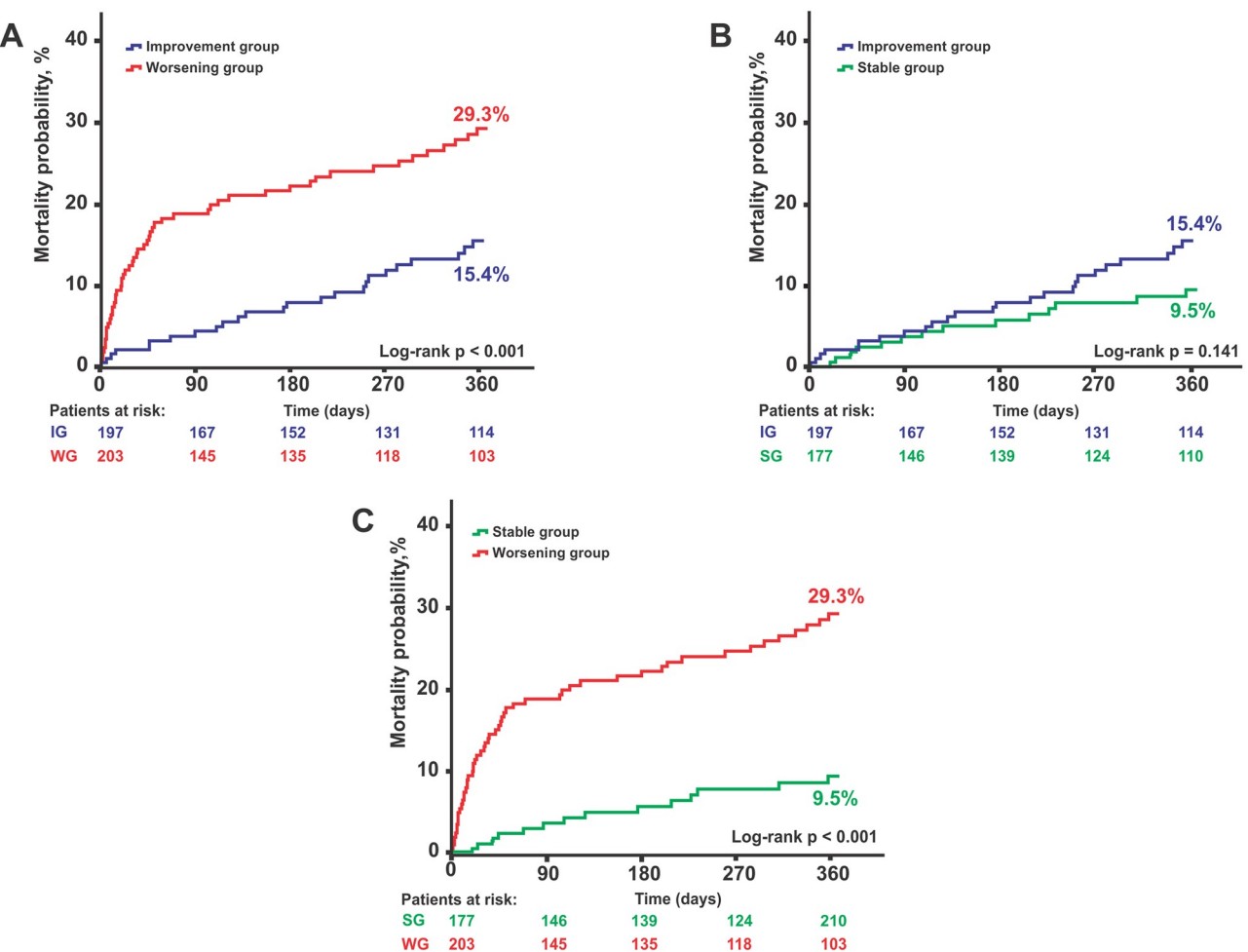

**Fig 2. Kaplan-Meier curves for 1-year all-cause mortality.** Comparison of TIRFI group vs. AKI group (A), TIRFI group vs. stable renal function group (B) and stable renal function group vs. AKI group (C) 1-Year all-cause mortality rates. Abbreviations: AKI indicates acute kidney injury; SRFG indicates stable renal function group; TIRFI indicates TAVR induced renal function improvement.

procedure [10, 11, 22]. Beyond that, the high proportion of CKD among these patients, ranging from 52% to 72% in previous large studies [10, 11, 21] reflects the impact of these comorbidities. In our study, CKD was present in 577 (70%) of the patients from the Brazilian TAVR Registry, with an average age of 80 years and high-surgical risk according STS score.

An interesting point of our study is that TIRFI occurred similarly compared with those with AKI or stable renal function (34.1% vs. 35.2% vs. 30.7%, respectively), despite all the adverse procedural features, such as the use of contrast media, atherosclerotic particles renal embolization during catheter passage through the aorta and deployment of the valve prosthesis [23], and also the occurrence of low cardiac output during rapid ventricle pacing for balloon expandable prosthesis deployment or pre- or post-dilation of the aortic valve, which was associated with acute kidney injury and short- and long-term mortality if multiple ($\geq$ 3 episodes) or prolonged rapid ventricular pacing duration ($\geq$ 36 seconds) are performed [24].

Attempting to determine the predictors of TIRFI, we found that the absence of coronary artery disease and lower eGFR were independent predictors for renal function recovery after TAVR. Regarding to eGFR as a predictor of TIRFI, the lower the eGFR, the greater is the

**Table 3. Predictors of TIRFI after TAVR procedure.**

| Variable | Univariable Analysis | | Multivariable Analysis | |
|---|---|---|---|---|
| | OR (95% CI) | P value | OR (95% CI) | P value |
| **Age, years** | 0.98 (0.95–1.00) | 0.13 | - | - |
| **Male sex** | 0.84 (0.59–1.19) | 0.32 | - | - |
| **NYHA class III/IV** | 1.11 (0.69–1.76) | 0.67 | - | - |
| **Diabetes** | 0.76 (0.52–1.11) | 0.15 | - | - |
| **Hypertension** | 1.09 (0.74–1.63) | 0.65 | - | - |
| **COPD** | 1.01 (0.65–1.56) | 0.98 | - | - |
| **Pulmonary hypertension** | 0.77 (0.51–1.16) | 0.20 | - | - |
| **CAD** | 0.73 (0.51–1.04) | **0.077** | 0.69 (0.48–0.98) | **0.039** |
| **Peripheral vascular disease** | 1.13 (0.72–1.78) | 0.60 | - | - |
| **Previous CABG** | 0.96 (0.62–1.46) | 0.83 | - | - |
| **STS score, %** | 1.00 (0.98–1.03) | 0.45 | - | - |
| **eGFR, mL/min/1.73m$^2$** | 0.98 (0.96–0.99) | **0.014** | 0.98 (0.97–1.00) | **0.008** |
| **Diuretics** | 1.15 (0.80–1.64) | 0.46 | - | - |
| **ACE inhibitors or ARB** | 1.27 (0.90–1.79) | 0.18 | - | - |
| **Beta-blockers** | 1.06 (0.75–1.51) | 0.73 | - | - |
| **Statin** | 0.96 (0.68–1.37) | 0.83 | - | - |
| **LVEF, %** | 0.99 (0.98–1.00) | 0.45 | - | - |
| **Mean transaortic gradient, mmHg** | 1.00 (0.99–1.02) | 0.10 | - | - |
| **AVA, cm$^2$** | 1.68 (0.64–4.38) | 0.28 | - | - |
| **Contrast media volume, mL** | 1.00 (0.99–1.00) | 0.71 | - | - |
| **Procedure access (except Transfemoral)** | 0.58 (0.27–1.25) | 0.16 | - | - |
| **Prosthetic valve type** | - | 0.55 | - | - |
| Corevalve | 1.00 | - | - | - |
| Sapien XT | 1.00 (0.67–1.51) | - | - | - |
| Inovare prosthesis | 0.54 (0.18–1.68) | - | - | - |
| **Period of TAVR procedure** | | 0.25 | - | - |
| T1 (2008–2010) | 1.00 | - | - | - |
| T2 (2011–2013) | 0.87 (0.54–1.38) | - | - | - |
| T3 (2014–2015) | 1.24 (0.72–2.14) | - | - | - |

95% CI, 95% confidence interval.

Abbreviations: ACE indicates angiotensin-converting enzyme; ARB, angiotensin receptor blocker; AVA, aortic valve area; CABG, coronary artery bypass graft; CAD, coronary artery disease; COPD, chronic obstructive pulmonary disease; eGFR, estimated glomerular filtration rate; LVEF, left ventricular ejection fraction; NYHA, New York Heart Association; OR, odds ratio; STS, Society of Thoracic Surgeons; TAVR, transcatheter aortic valve replacement; TIRFI, TAVR induced renal function improvement.

chance of improvement in renal function. It can suggest that improvement in renal function is primarily due to the hemodynamics effects after TAVR procedure to relief aortic stenosis, increasing the cardiac output and leading to a better renal perfusion, suggesting that a mechanism of type 2 cardiorenal syndrome may be involved [25]. In the present analysis, there was no difference between the 3 groups related to contrast volume, use of diuretics and other variables that may contribute to deterioration of renal function. Importantly, the baseline eGFR was slightly lower among patients in the TIRFI group when compared with the AKI group, corroborating for the hypothesis that low cardiac output may be the main mechanism of impaired renal function at baseline in these population and explain why the presence of lower eGFR was found as an independent predictor of TIRFI. Also, patients without coronary artery

**Table 4. Predictors of AKI after TAVR procedure.**

| Variable | Univariable Analysis | | Multivariable Analysis | |
|---|---|---|---|---|
| | OR (95% CI) | P value | OR (95% CI) | P value |
| **Age, year** | 1.02 (0.99–1.05) | **0.06** | 1.03 (1.00–1.05) | **0.033** |
| **Male sex** | 0.83 (0.59–1.17) | 0.29 | - | - |
| **NYHA class III/IV** | 0.99 (0.63–1.56) | 0.95 | - | - |
| **Diabetes** | 0.17 (0.82–1.69) | 0.38 | - | - |
| **Hypertension** | 1.71 (0.13–2.59) | **0.011** | 1.69 (1.11–2.57) | **0.014** |
| **COPD** | 0.90 (0.58–1.40) | 0.65 | | |
| **Pulmonary hypertension** | 0.94 (0.63–1.40) | 0.75 | - | - |
| **CAD** | 0.99 (0.69–1.40) | 0.93 | - | - |
| **Peripheral vascular disease** | 1.07 (0.68–1.69) | 0.77 | - | - |
| **Previous CABG** | 0.74 (0.48–1.15) | 0.18 | - | - |
| **STS score, %** | 1.00 (0.97–1.02) | 0.98 | - | - |
| **eGFR, mL/min/1.73m$^2$** | 1.01 (0.99–1.02) | **0.008** | 1.01 (1.00–1.02) | 0.053 |
| **Diuretics** | 1.15 (0.80–1.64) | 0.44 | - | - |
| **ACE inhibitors or ARB** | 1.11 (0.79–1.57) | 0.54 | - | - |
| **Beta-blockers** | 0.93 (0.65–1.34) | 0.66 | - | - |
| **Statin** | 1.22 (0.86–1.74) | 0.26 | - | - |
| **LVEF, %** | 1.00 (0.99–1.01) | 0.28 | - | - |
| **Mean transaortic gradient, mmHg** | 1.00 (0.99–1.01) | 0.97 | - | - |
| **AVA, cm$^2$** | 0.71 (0.26–1.87) | 0.49 | - | - |
| **Contrast media volume, mL** | 0.99 (0.99–1.00) | 0.60 | - | - |
| **Non-transfemoral access** | 0.93 (1.00–3.73) | **0.048** | 2.07 (1.06–4.07) | **0.034** |
| **Prosthetic valve type** | - | 0.62 | - | - |
| **Corevalve** | 1.00 | - | - | - |
| **Sapien XT** | 0.90 (0.60–1.36) | - | - | - |
| **Inovare prosthesis** | 1.46 (0.56–3.77) | - | - | - |
| **Period of TAVR procedure** | | 0.23 | - | - |
| T1 (2008–2010) | 1.00 | - | - | - |
| T2 (2011–2013) | 0.73 (0.47–1.15) | - | - | - |
| T3 (2014–2015) | 0.63 (0.36–1.08) | - | - | - |

95% CI, 95% confidence interval.

Abbreviations: ACE indicates angiotensin-converting enzyme; AKI, acute kidney injury; ARB, angiotensin receptor blocker; AVA, aortic valve area; CABG, coronary artery bypass graft; CAD, coronary artery disease; COPD, chronic obstructive pulmonary disease; eGFR, estimated glomerular filtration rate; LVEF, left ventricular ejection fraction; NYHA, New York Heart Association; OR, odds ratio; STS, Society of Thoracic Surgeons. TAVR, transcatheter aortic valve replacement.

disease may have less peripheral artery disease incidence and lower rates of atherosclerotic emboli during the TAVR procedure, and that may partially explain the association with TIRFI.

The presence of hypertension and higher age were associated with AKI, and the endothelial dysfunction added to the occurrence of low cardiac output during the procedure could be more harmful for this population. Furthermore, the use of non-transfemoral access sites was associated with AKI after TAVR, as well as AKI group had also more procedure complications such as valve malpositioning, major or life-threatening bleeding and major vascular complication. Also, the higher eGFR at baseline, the greater is the chance of developing AKI after TAVR. This finding was probably related simply to the percentage variation of eGFR in a

**Table 5. Multivariable predictors of 1-year all-cause mortality.**

| Variable | Univariable Analysis | | Multivariable Analysis | |
|---|---|---|---|---|
| | Hazard Ratio (95% CI) | P value | Hazard Ratio (95% CI) | P value |
| Age, years* | 1.05 (0.90–1.22) | 0.50 | - | - |
| Male sex | 0.77 (0.51–1.15) | 0.21 | - | - |
| NYHA class III/IV | 2.16 (1.04–4.45) | **0.038** | 1.74 (0.83–3.63) | 0.14 |
| Diabetes | 1.46 (0.97–2.22) | 0.07 | - | - |
| Hypertension | 1.45 (0.88–2.41) | 0.14 | - | - |
| COPD | 2.14 (1.38–3.32) | **0.001** | 1.72 (1.06–2.79) | **0.028** |
| Pulmonary hypertension | 1.69 (1.09–2.60) | **0.017** | 1.36 (0.86–2.16) | 0.18 |
| CAD | 1.30 (0.85–1.99) | 0.23 | - | - |
| Peripheral vascular disease | 1.89 (1.18–3,04) | **0.008** | 1.58 (0.92–2.69) | 0.09 |
| Previous CABG | 1.07 (0.65–1.76) | 0.78 | - | - |
| STS score, % | 1.04 (1.02–1.06) | **0.001** | - | - |
| Baseline eGFR, mL/min/1.73m²** | 0.69 (0.59–0.81) | **< 0.001** | 0.73 (0.61–0.86) | **< 0.001** |
| Baseline eGFR < 30 mL/min/1.73m² | 2.02 (1.33–3.09) | **0.001** | - | - |
| Diuretics | 1.46 (0.93–2.30) | 0.09 | - | - |
| ACE inhibitors or ARB | 0.80 (0.54–1.21) | 0.30 | - | - |
| Beta-blockers | 0.79 (0.50–1.19) | 0.25 | - | - |
| Statin | 0.97 (0.64–1.48) | 0.91 | - | - |
| LVEF, %** | 0.92 (0.81–1.37) | 0.18 | - | - |
| Mean transaortic gradient, mmHg** | 0.94 (0.82–1.07) | 0.34 | - | - |
| AVA, cm² | 0.51 (0.15–1.73) | 0.28 | - | - |
| Contrast media volume, mL | 1.09 (0.87–1.37) | 0.44 | - | - |
| Procedure access (except Transfemoral) | 2.34 (1.25–4.39) | **0.008** | 1.29 (0.53–3.17) | 0.57 |
| Sapien XT prosthesis | 0.76 (0.44–1.31) | 0.32 | - | - |
| Inovare prosthesis | 2.55 (1.03–6.34) | **0.043** | - | - |
| Myocardial infarction | 4.47 (1.09–18.09) | **0.037** | - | - |
| Stroke/TIA | 3.96 (2.34–6.71) | **< 0.001** | 3.08 (1.74–5.44) | **< 0.001** |
| Major or life-threatening bleeding | 1.72 (0.89–3.32) | 0.10 | - | - |
| Major vascular complication | 1.80 (1.03–3.13) | **0.037** | 1.25 (0.69–2.25) | 0.45 |
| New LBBB | 1.03 (0.66–1.61) | 0.89 | - | - |
| AV block | 0.85 (0.52–1.38) | 0.51 | - | - |
| Valve malpositioning | 4.35 (2.31–8.18) | **< 0.001** | 1.94 (0.92–4.07) | 0.08 |
| New pacemaker | 0.88 (0.53–1.38) | 0.38 | - | - |
| TIRFI | 1.40 (0.84–3.15) | 0.14 | 1.45 (0.75–2.83) | 0.27 |
| AKI | 3.82 (2.12–6.88) | **< 0.001** | 3.42 (1.87–6.23) | **< 0.001** |

95% CI, 95% confidence interval.

Abbreviations: ACE indicates angiotensin-converting enzyme; AKI, acute kidney injury; ARB, angiotensin receptor blocker; AV, atrioventricular; AVA, aortic valve area; CABG, coronary artery bypass graft; CAD, coronary artery disease; COPD, chronic obstructive pulmonary disease; eGFR, estimated glomerular filtration rate; HR, hazard ratio; LBBB, left bundle branch block; LVEF, left ventricular ejection fraction; NYHA, New York Heart Association; STS, Society of Thoracic Surgeons; TIA, transient ischemic attack; TIRFI, TAVR induced renal function improvement.

* For each increase of 5 units in age.

** For each increase of 10 units in baseline eGFR, LVEF or mean transaortic gradient.

*** For each increase of 100 units in contrast media volume.

high-risk population with multiple comorbidities, since the results of the statistical analysis were borderline.

According to previous studies, AKI was associated with poor outcomes, including higher 1-year all-cause and cardiovascular mortality rates [9, 16, 26]. However, our study did not show association between TIRFI and lower 30-days and 1-year all-cause mortality or other outcomes when compared with stable renal function group. Our results also sustain the findings from the subanalysis of Partner 1 which included a similar population and showed no reduction in mortality rates and other outcomes in the TIRFI group [9].

A recent single-center prospective study showed that patients who had TIRFI after TAVR may also have better 30-day and 2-year outcomes, independently of the baseline kidney function [16]. The reason for discordance of our findings with this recent study could be in part because our study population surgical risk was slightly higher using the STS score, which may justify an increase in mortality even in the stable renal function and TIRFI groups.

## Limitations

This is a prospective observational multicenter study using the Brazilian TAVR Registry and there are several inherent limitations. First, data was self-reported by each center and on-site source document verification was randomly performed in only 20% of cases. Registries studies usually do not perform source document verification, and therefore, we consider that 20% of random verification is a representative sample to validity of the data. Besides that, there is no data about measure of urine output, pre- and post-hydration strategies and proteinuria levels that might be important for AKI definition. Since there is no definition of TIRFI, we arbitrary used variation in eGFR between baseline and at discharge after TAVR of 10% for both TIRFI and AKI criteria. As most of the patients had high-surgical risk using STS score, we could not extrapolate our results for intermediate- and low-surgical risk patients. Also, despite coronary angiography are usually performed before admission for TAVR, as a routine, for evaluation of concomitant coronary artery disease in Brazilian centers which included patients in the Brazilian TAVR Registry, information on how many patients had coronary angiography and percutaneous coronary intervention during the admission for TAVR was not available. Finally, we had around 27% loss of follow-up at 1 year. However, the median follow-up of the patients included in the present study was 385 [162–742] days, and therefore, we have chosen to assess the outcomes at 1 year.

## Conclusion

In our study, TIRFI was frequently found in patients with prior impaired renal function and promoted lower mortality when compared with patients with AKI. The absence of coronary artery disease and lower eGFR at baseline were independent predictors of TIRFI.

## Supporting information

**S1 Table. Baseline and procedural characteristics of patients with loss of follow-up versus patients with complete 1-year follow-up.**
(DOCX)

**S2 Table. Predictors of TIRFI after TAVR procedure (including outcomes).**
(DOCX)

**S3 Table. Predictors of AKI after TAVR procedure (including outcomes).**
(DOCX)

**S1 File.**
(XLSX)

## Acknowledgments

The authors acknowledge the expert work of Rogério R. Prado for the statistical support.

## Author Contributions

**Conceptualization:** Michel V. Lemes da Silva, Antonio C. B. Nunes Filho, Vitor E. E. Rosa, Adriano Caixeta, Pedro A. Lemos Neto, Henrique B. Ribeiro, Breno O. Almeida, José Mariani, Jr, Carlos M. Campos, Alexandre A. C. Abizaid, Roney O. Sampaio, Paulo Caramori, Rogério Sarmento-Leite, Flávio Tarasoutchi, Marcelo Franken, Fábio S. de Brito, Jr.

**Data curation:** Adriano Caixeta, Pedro A. Lemos Neto, Henrique B. Ribeiro, Breno O. Almeida, José Mariani, Jr, Carlos M. Campos, Alexandre A. C. Abizaid, José A. Mangione, Paulo Caramori, Rogério Sarmento-Leite, Fábio S. de Brito, Jr.

**Formal analysis:** Michel V. Lemes da Silva, Antonio C. B. Nunes Filho, Vitor E. E. Rosa, Adriano Caixeta, Pedro A. Lemos Neto, Henrique B. Ribeiro, Carlos M. Campos, Roney O. Sampaio, Flávio Tarasoutchi, Marcelo Franken, Fábio S. de Brito, Jr.

**Funding acquisition:** Adriano Caixeta, Pedro A. Lemos Neto, Henrique B. Ribeiro, Breno O. Almeida, José Mariani, Jr, Carlos M. Campos, Alexandre A. C. Abizaid, José A. Mangione, Paulo Caramori, Fábio S. de Brito, Jr.

**Investigation:** Adriano Caixeta, Pedro A. Lemos Neto, Henrique B. Ribeiro, Breno O. Almeida, José Mariani, Jr, Carlos M. Campos, Alexandre A. C. Abizaid, José A. Mangione, Paulo Caramori, Rogério Sarmento-Leite, Fábio S. de Brito, Jr.

**Methodology:** Michel V. Lemes da Silva, Antonio C. B. Nunes Filho, Vitor E. E. Rosa, Pedro A. Lemos Neto, Henrique B. Ribeiro, Carlos M. Campos, Alexandre A. C. Abizaid, Roney O. Sampaio, Paulo Caramori, Rogério Sarmento-Leite, Fábio S. de Brito, Jr.

**Project administration:** Michel V. Lemes da Silva, Antonio C. B. Nunes Filho, Adriano Caixeta, Henrique B. Ribeiro, Breno O. Almeida, José Mariani, Jr, Alexandre A. C. Abizaid, José A. Mangione, Fábio S. de Brito, Jr.

**Resources:** Adriano Caixeta, Pedro A. Lemos Neto, José Mariani, Jr, Carlos M. Campos, Alexandre A. C. Abizaid, José A. Mangione, Paulo Caramori, Rogério Sarmento-Leite, Fábio S. de Brito, Jr.

**Software:** Alexandre A. C. Abizaid, Fábio S. de Brito, Jr.

**Supervision:** Antonio C. B. Nunes Filho, Vitor E. E. Rosa, Fábio S. de Brito, Jr.

**Validation:** Adriano Caixeta, Pedro A. Lemos Neto, Henrique B. Ribeiro, Breno O. Almeida, José Mariani, Jr, Carlos M. Campos, Alexandre A. C. Abizaid, Roney O. Sampaio, Paulo Caramori, Rogério Sarmento-Leite, Flávio Tarasoutchi, Marcelo Franken, Fábio S. de Brito, Jr.

**Visualization:** Michel V. Lemes da Silva, Antonio C. B. Nunes Filho, Adriano Caixeta, Henrique B. Ribeiro, José Mariani, Jr, Carlos M. Campos, Alexandre A. C. Abizaid, José A. Mangione, Roney O. Sampaio, Paulo Caramori, Rogério Sarmento-Leite, Flávio Tarasoutchi, Marcelo Franken, Fábio S. de Brito, Jr.

**Writing – original draft:** Michel V. Lemes da Silva, Antonio C. B. Nunes Filho, Vitor E. E. Rosa, Adriano Caixeta, Pedro A. Lemos Neto, Henrique B. Ribeiro, Breno O. Almeida, José Mariani, Jr, Carlos M. Campos, Alexandre A. C. Abizaid, José A. Mangione, Roney O. Sampaio, Paulo Caramori, Rogério Sarmento-Leite, Flávio Tarasoutchi, Marcelo Franken, Fábio S. de Brito, Jr.

**Writing – review & editing:** Michel V. Lemes da Silva, Antonio C. B. Nunes Filho, Vitor E. E. Rosa, Adriano Caixeta, Pedro A. Lemos Neto, Breno O. Almeida, José Mariani, Jr, Carlos M. Campos, Alexandre A. C. Abizaid, José A. Mangione, Roney O. Sampaio, Paulo Caramori, Rogério Sarmento-Leite, Flávio Tarasoutchi, Marcelo Franken, Fábio S. de Brito, Jr.

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
