## [Decision Letter · Decision Letter 0]

22 Jan 2021

PONE-D-20-31591

Improvement of renal function after transcatheter aortic valve replacement in patients with chronic kidney disease

PLOS ONE

Dear Dr. Antonio C. B. Nunes Filho,

Thank you for submitting your manuscript to PLOS ONE. After careful consideration, we feel that it has merit but does not fully meet PLOS ONE’s publication criteria as it currently stands. Therefore, we invite you to submit a revised version of the manuscript that addresses the points raised during the review process.

We look forward to receiving your revised manuscript.

Kind regards,

Ping-Hsun Wu, M.D. PhD.

Academic Editor

PLOS ONE

Additional Editor Comments:

The time interval between the first and second renal function records should be mentioned. The information on subjects with loss follow-up needed to be addressed in the study. An extended period of enrollment in this study, so controlling this potential confounding factor could be considered in the analysis process.

"I have read the journal's policy and the authors of this manuscript have the following competing interests:

Dr. de Brito Jr. and Dr. Mangione are proctors for Edwards Lifescience and Medtronic. Dr. Ribeiro is proctor and consultant for Edwards Lifescience, Medtronic and Boston Scientific. Dr. Caramori is proctor for Medtronic. All other authors have reported that they have no relationships relevant to the contents of this paper to disclose."

Reviewers' comments:

Reviewer's Responses to Questions

**Comments to the Author**

1. Is the manuscript technically sound, and do the data support the conclusions?

Reviewer #1: Yes

Reviewer #2: Yes

2. Has the statistical analysis been performed appropriately and rigorously? 

Reviewer #1: No

Reviewer #2: Yes

3. Have the authors made all data underlying the findings in their manuscript fully available?

Reviewer #1: Yes

Reviewer #2: No

4. Is the manuscript presented in an intelligible fashion and written in standard English?

Reviewer #1: Yes

Reviewer #2: Yes

5. Review Comments to the Author

Reviewer #1: The current study by de Silva et al. explores the impact of TAVR for severe AS on renal function. The authors report that of their total patient population undergoing TAVI from 2008-2015, 70% had CKD at baseline. Of these, after undergoing TAVI, about 1/3 patients had an improvement in renal function, 1/3 had worsening renal function, and the remaining 1/3 had stable renal function. The authors also note in their analysis the predictors of TIRFI and AKI post-TAVI, as well predictors of 30-day and 1-year mortality.

Comments:

1. The authors should not mention p values derived from 2 different comparisons within the same column as they do in Tables 1 and 2. This is confusing for readers to interpret. If they want to show results from both comparisons, they should put the p values in 2 different columns specifying the comparison.

2. Similarly, the authors should cross check the p value for correction for eGFR between TIRFI vs stable renal group as the numbers are different in the results text and table 1.

3. According to the tables, baseline eGFR was a predictor for both TIRFI and AKI post- TAVR. The authors must provide more clarification regarding this in their text.

4. Can the authors clarify how the data regarding follow-up was collected for their patients? Furthermore, can the authors provide information regarding patients lost to follow-up? Was data available for all subjects up to 1 year?

5. The authors state that source document verification was performed only in 20% of the patients. Can the authors clarify this further as this may have impact on the results and validity of the data

6. Furthermore, according to the data, these patients underwent TAVI procedures over a time period of 7 years, during which the techniques of the interventional procedure and device structure changed considerably. How do the authors account for this in their results?

Reviewer #2: This is multi-center prospective registration study to investigate the predictors of improvement of renal function after transcatheter aortic valve replacement (TAVR) by Lemes da Silva et al. The authors reported that 34.1% of patients presented with improvement of renal function after transcatheter aortic valve replacement. Absence of coronary artery disease and lower baseline eGFR were independent predictors of improvement of renal function after TAVR. However, TAVR induced renal function improvement was not associated with improved 1-year outcomes.

Comments and questions for the authors are as follows:

1. When was the second time point for determination of renal function(eGFR)? What was the time interval between the first and second renal function? Was there a standard protocol?

2. How many patients had coronary angiography during the admission for TAVR? Did patients with coronary artery disease undergo percutaneous coronary intervention during the admission for TAVR?

3. There were high procedure complications in CKD group. Did the authors put these variables in the analysis of predictors of CKD?

4. Page 6 line 124 “leak aortic jet velocity of ≥ 4.0 m/s” should be peak aortic jet velocity of ≥ 4.0 m/s.

6. PLOS authors have the option to publish the peer review history of their article (what does this mean?). If published, this will include your full peer review and any attached files.

Reviewer #1: No

Reviewer #2: No

---

## [Author Response · Author response to Decision Letter 0]

24 Mar 2021

Dear, Ping-Hsun Wu, M.D. PhD., Academic Editor, PLOS ONE. We thank for the positive comments and suggestions. We have implemented your suggestions in the revised manuscript and we hope that the new version will be to your satisfaction.

Response to Reviewers 

Thank you for submitting your manuscript to PLOS ONE. After careful consideration, we feel that it has merit but does not fully meet PLOS ONE’s publication criteria as it currently stands. Therefore, we invite you to submit a revised version of the manuscript that addresses the points raised during the review process.

 - We thank the editor and reviewers for the positive comments on our work and the constructive suggestions that have contributed to the improvement of our manuscript. We have implemented your suggestions in the revised manuscript and we hope that the new version will be to your satisfaction. We are open to any additional suggestions.

1. The time interval between the first and second renal function records should be mentioned.

 - The editor made a very important point. Following the recommendation, we described more clearly the time between the first and the second renal function record (Page 6, Lines 138-142). Therefore, we highlight that the first creatinine (baseline) was collected on the day or the day before TAVR procedure, and the second creatinine used to calculate variation on renal function was at discharge. Moreover, the median length of stay was 7 days (1-368) and the mean hospitalization period was 13 days.

2. The information on subjects with loss follow-up needed to be addressed in the study.

 - We thank the editor and found the suggestion extremely pertinent. Thus, we included a Follow-up section on Results section describing the information on subjects with loss of follow-up (Page 13, Lines 237-243). Also, we compared baseline characteristic of patients with loss of follow-up with those who completed followed up in 1 year. The comparison of baseline characteristics was highlighted in Supplemental Table 1. The follow-up was performed by phone calls at 1 month, 1 year and then annually (Page 5, Lines 114-115). We had around 27% loss of follow-up at 1 year, however, the median follow-up of our study was 385 [162 – 742] days, and therefore, we have chosen to assess the outcomes at 1 year. This practice is already well recognized and was performed in large studies like EXCEL Trial (N Engl J Med 2016;375:2223-35), in which the authors used the median follow-up in 3 years to assess the primary composite endpoint of the study. We have highlighted this issue as a potential limitation on appropriate section (Pages 22-23, Lines 380-382).

3. An extended period of enrollment in this study, so controlling this potential confounding factor could be considered in the analysis process.

 - The Editors raised an important aspect. Following the suggestion, we have divided the study population into tertiles according to the year they were submitted to TAVR procedure. This information is included in Table 1 and throughout the manuscript (Page 9, Lines 196-200): 100 patients underwent TAVR from 2008 to 2010(T1); 352 patients from 2011 to 2013 (T2); and 125 from 2014 to 2015 (n = 125) –(T3), with no statistical differences between the 3 tertiles (p = 0.15). However, neither tertiles was predictor for either TAVR induced renal function improvement (TIRFI) (p = 0.25) or Acute Kidney Injury (AKI) (p = 0.23), as shown in Tables 3 and 4.

Reviewer #1: The current study by de Silva et al. explores the impact of TAVR for severe AS on renal function. The authors report that of their total patient population undergoing TAVI from 2008-2015, 70% had CKD at baseline. Of these, after undergoing TAVI, about 1/3 patients had an improvement in renal function, 1/3 had worsening renal function, and the remaining 1/3 had stable renal function. The authors also note in their analysis the predictors of TIRFI and AKI post-TAVI, as well predictors of 30-day and 1-year mortality.

 - We thank the Reviewer for his/her comments and for the constructive suggestions that have contributed to improving our work. 

1. The authors should not mention p values derived from 2 different comparisons within the same column as they do in Tables 1 and 2. This is confusing for readers to interpret. If they want to show results from both comparisons, they should put the p values in 2 different columns specifying the comparison.

 - The Reviewer made a good suggestion. We highlighted that the p value described is from comparison within the 3 groups using legends “a”, and legends “b” and “c” were used to describe difference between stable vs AKI groups and TIRFI vs AKI groups, respectively (Tables 1 and 2). 

2. Similarly, the authors should cross check the p value for correction for eGFR between TIRFI vs stable renal group as the numbers are different in the results text and table 1.

 - The Reviewer is right. We have corrected this information throughout the manuscript accordingly. We also informed P value for statistical difference between the 3 groups and the P value for the post-hoc analysis describing which groups the difference was related to (Page 8, Lines 190-191). The same has been done for Hypertension (Page 8, Lines 193-194).

3. According to the tables, baseline eGFR was a predictor for both TIRFI and AKI post- TAVR. The authors must provide more clarification regarding this in their text.

 - The Reviewer made an important point. We have highlighted the potential mechanism throughout the “Discussion” section (as stated on Page 21, Lines 337-340). As previously described by Ronco et al (Eur Heart J 2010; 31(6):703-11), any chronic cardiac abnormality that could lead to a low cardiac output condition and chronic renal hypoperfusion may impact on renal function. The relief of aortic stenosis after TAVR could improve the hemodynamics and lead to a better renal perfusion, if this is the main condition leading to CKD. In our study, the eGFR was lower among patients in the TIRFI group, when compared to AKI group. It is possible that the patients in the TIRFI group may have benefited more of TAVR procedure due to worse hemodynamic condition caused by the aortic stenosis and, therefore, explain why the lower the eGFR is, the greater is the chance of TIRFI (as highlighted in Page 14, Line 262; and Pages 20-21, Lines 330-331). On the other hand, higher eGFR was a predictor of AKI in multivariable analysis, as shown in Table 4 (HR: 1.01; 95% CI, 1.00 – 1.02; P = 0.053). However, this result was borderline as shown by the results of our analysis, and was probably related simply to percentage variation of eGFR in a high-risk population with multiple comorbidities (Page 14, Lines 265-268; and Page 21, Lines 349-352).

4. Can the authors clarify how the data regarding follow-up was collected for their patients? Furthermore, can the authors provide information regarding patients lost to follow-up? Was data available for all subjects up to 1 year?

 - The Reviewer raised an important point. The follow-up was performed by phone calls at 1 month, 1 year and then annually (Page 5, Lines 114-115). We included a Follow-up section on Results describing the information on subjects with loss of follow-up (Pages 13, Lines 237-243). Also, we compared baseline characteristic of patients with loss of follow-up with those who completed followed up in 1 year. The comparison of baseline characteristics was highlighted in a Supplemental Table 1. We had around 27% loss of follow-up at 1 year, however, the median follow-up of our study was 385 [162 – 742] days, and therefore, we have chosen to assess the outcomes at 1 year. This practice is already well recognized and was performed in large studies like EXCEL Trial (N Engl J Med 2016;375:2223-35), in which the authors used the median follow-up in 3 years to assess the primary composite endpoint of the study. We have highlighted this issue as a potential limitation on appropriate section (Pages 22-23, Lines 380-382).

5. The authors state that source document verification was performed only in 20% of the patients. Can the authors clarify this further as this may have impact on the results and validity of the data?

 - The Reviewer raised an interesting point. The data verification was randomly performed in one-fifth of the patients included in the Brazilian TAVR Registry, as determined and allowed by our ethical committee, and was highlighted as a potential limitation of our study. Registries studies usually do not perform source document verification, and therefore, we consider that 20% of random verification is a representative sample to validity of the data (Page 22, Lines 369-370).

6. Furthermore, according to the data, these patients underwent TAVI procedures over a time period of 7 years, during which the techniques of the interventional procedure and device structure changed considerably. How do the authors account for this in their results?

 - The Reviewer made an important point. Following the Reviewer and Editor’s suggestion, we have divided the study population into tertiles according to the year they were submitted to TAVR procedure. This information is included in Table 1 and throughout the manuscript (Page 9, Lines 196-200): 100 patients underwent TAVR from 2008 to 2010 (T1); 352 patients from 2011 to 2013 (T2); and 125 from 2014 to 2015 (T3), with no statistical differences between the 3 tertiles (p = 0.15). However, neither tertiles was predictor for either TAVR induced renal function improvement (TIRFI) (p = 0.25) or Acute Kidney Injury (AKI) (p = 0.23), as shown in Tables 3 and 4.

 

Reviewer #2: This is multi-center prospective registration study to investigate the predictors of improvement of renal function after transcatheter aortic valve replacement (TAVR) by Lemes da Silva et al. The authors reported that 34.1% of patients presented with improvement of renal function after transcatheter aortic valve replacement. Absence of coronary artery disease and lower baseline eGFR were independent predictors of improvement of renal function after TAVR. However, TAVR induced renal function improvement was not associated with improved 1-year outcomes.

 - We thank the Reviewer for his/her comments and for the constructive suggestions that have contributed to improving our work. 

1. When was the second time point for determination of renal function (eGFR)? What was the time interval between the first and second renal function? Was there a standard protocol?

 - The editor made a very important point. Following the recommendation, we described more clearly the time between the first and the second renal function record (Page 6, Lines 138-142). Therefore, we highlight that the first creatinine (baseline) was collected on the day or the day before TAVR procedure, and the second creatinine used to calculate variation on renal function was at discharge. Moreover, the median length of stay was 7 days (1-368) and the mean hospitalization period was 13 days.

2. How many patients had coronary angiography during the admission for TAVR? Did patients with coronary artery disease undergo percutaneous coronary intervention during the admission for TAVR?

 - This is an important issue raised by the Reviewers. The coronary angiography are usually performed before admission for TAVR, as a routine, for evaluation of concomitant coronary artery disease in all Brazilian centers which included patients in the Brazilian TAVR Registry. However, data on how many patients had coronary angiography and percutaneous coronary intervention during the admission for TAVR was not available. The only data available is that 32 (5.5%) of the patients have undergone percutaneous coronary intervention during TAVR procedure. We have included the lack of data about this subject on “Limitations” section (Page 22, Lines 376-380).

3. There were high procedure complications in CKD group. Did the authors put these variables in the analysis of predictors of CKD?

 - The Reviewer raised a very important point. At first, our primary objective was to evaluate which clinical baseline characteristics could predict the improvement and worsening in renal function. Based on that, we could evaluate the risk of our patients develop worsening in renal function after TAVR procedure, as well as take stricter measures to minimize the impact of this condition. However, following the recommendation of the Reviewer, we made a new model including procedure complications as variables in both analyses of predictors of TIRFI and AKI, as supplemental tables 2 and 3. Yet, we have stated these supplemental materials in the revised manuscript (Page 14, Lines 265-268)

4. Page 6 line 124 “leak aortic jet velocity of ≥ 4.0 m/s” should be peak aortic jet velocity of ≥ 4.0 m/s.

 - We the reviewer is correct. The typo was corrected (Page 6, Line 124).

---

## [Decision Letter · Decision Letter 1]

20 Apr 2021

Improvement of renal function after transcatheter aortic valve replacement in patients with chronic kidney disease

PONE-D-20-31591R1

Dear Dr. Antonio C. B. Nunes Filho,

We’re pleased to inform you that your manuscript has been judged scientifically suitable for publication and will be formally accepted for publication once it meets all outstanding technical requirements.

Kind regards,

Ping-Hsun Wu, M.D. PhD.

Academic Editor

PLOS ONE

**Comments to the Author**

1. If the authors have adequately addressed your comments raised in a previous round of review and you feel that this manuscript is now acceptable for publication, you may indicate that here to bypass the “Comments to the Author” section, enter your conflict of interest statement in the “Confidential to Editor” section, and submit your "Accept" recommendation.

Reviewer #2: All comments have been addressed

2. Is the manuscript technically sound, and do the data support the conclusions?

Reviewer #2: Yes

3. Has the statistical analysis been performed appropriately and rigorously? 

Reviewer #2: Yes

4. Have the authors made all data underlying the findings in their manuscript fully available?

Reviewer #2: Yes

5. Is the manuscript presented in an intelligible fashion and written in standard English?

Reviewer #2: Yes

6. Review Comments to the Author

Reviewer #2: (No Response)

7. PLOS authors have the option to publish the peer review history of their article (what does this mean?). If published, this will include your full peer review and any attached files.

Reviewer #2: No

---

## [Editor Report · Acceptance letter]

4 May 2021

PONE-D-20-31591R1 

Improvement of renal function after transcatheter aortic valve replacement in patients with chronic kidney disease 

Dear Dr. Nunes Filho:

I'm pleased to inform you that your manuscript has been deemed suitable for publication in PLOS ONE. Congratulations! Your manuscript is now with our production department. 

Kind regards, 

on behalf of

Dr. Ping-Hsun Wu 

Academic Editor

PLOS ONE